# Do health professionals know about overdiagnosis in screening, and how are they dealing with it? A mixed-methods systematic scoping review

Veerle Piessens[1]*, Ann Van den Bruel[2], An Piessens[3], Ann Van Hecke[4], John Brandt Brodersen[5,6,7], Emelien Lauwerier[8], Florian Stul[1], An De Sutter[1], Stefan Heytens[1]

1 Department of Public Health and Primary Care, Faculty of Medicine and Health Sciences, Centre for Family Medicine, Ghent University, Ghent, Belgium, 2 Department of Public Health and Primary Care, Academic Centre for General Practice, KU Leuven, Leuven, Belgium, 3 Research Centre Pedagogy in Practice, KdG University of Applied Sciences and Arts, Antwerp, Belgium, 4 Department of Public Health and Primary Care, Faculty of Medicine and Health Sciences, University Centre for Nursing and Midwifery, Ghent University, Ghent, Belgium, 5 Department of Public Health, The Centre of General Practice, University of Copenhagen, Copenhagen, Denmark, 6 The Research Unit for General Practice in Region Zealand, Zealand, Denmark, 7 Department of Community Medicine, Faculty of Health Sciences, General Practice Research Unit, UiT, The Arctic University of Norway, Tromsø, Norway, 8 Department of Psychology, Open University Netherlands, Heerlen, the Netherlands

* Veerle.piessens@ugent.be

## Abstract

### Introduction

Medical screening is a major driver of overdiagnosis, which should be considered when making an informed screening decision. Health professionals (HPs) often initiate screening and are therefore responsible for informing eligible screening participants about the benefits and harms of screening. However, little is known about HPs' knowledge of overdiagnosis and whether they are prepared to inform screening candidates about this risk and enable people to make an informed screening decision.

### Methods

This is a systematic review of studies examining HPs' knowledge and perception of overdiagnosis, whether it affects their position on offering screening, and their willingness to inform screening candidates about overdiagnosis. We conducted systematic searches in MEDLINE, Embase, Web of Science, Scopus, CINAHL, and PsycArticles without language restrictions. Two authors analysed the qualitative and quantitative data separately. Confidence in the findings of the qualitative data was assessed using the GRADE-CERQual approach.

### Results

We included 23 publications after screening 9786 records. No studies directly examined HPs' *knowledge* of overdiagnosis. HPs' *perceptions* of overdiagnosis varied widely, from

**Data availability statement:** All relevant data are within the manuscript and its Supporting information files.

**Funding:** The author(s) received no specific funding for this work.

**Competing interests:** The authors have declared that no competing interests exist.

considering it a significant harm to seeing it as negligible. This seems linked to their overall beliefs about the benefits and harms of screening and to their position on offering screening, which varies from discouraging to actively promoting it. HPs also hold diverging approaches to informing screening candidates about overdiagnosis, from providing detailed explanations to limited or no information.

## Conclusion

There is a lack of research on HPs' knowledge of overdiagnosis, however, HPs who do know about overdiagnosis attribute substantially different levels of harm to it. This seems intertwined with their overall beliefs about the benefits of screening, their position towards offering screening, and their willingness to inform screening candidates about overdiagnosis. This has important implications for the public's right to evidence-based information and compromises an individual's right to make an informed screening decision.

## Introduction

Overdiagnosis in screening is the overdetection of an abnormal or pathological condition that would not have caused the person any harm if left undiscovered [1,2]. People who are overdiagnosed through screening cannot benefit from this early diagnosis but risk instead being harmed. First, the diagnosis itself is harmful because it labels a previously healthy person as a sick patient, causing fear, anxiety, and sometimes stigma, and can have implications for obtaining life or health insurance [3]. Second, most people with an overdetected condition will be overtreated because the very premise of screening is to catch a condition earlier to increase the chances of treatment success and improve the patient's prognosis. However, at the point of early diagnosis, it is impossible to differentiate between an overdiagnosed condition and a disease that would eventually become symptomatic later in life, leading to all screen-detected conditions being managed equally. However, treating a person with an overdiagnosed condition cannot provide any benefit but will cause harm through side effects of medication or radiation therapy, mutilating surgeries, risks of anaesthesia, pain, and discomfort [4–6]. Even deferred treatment, like active surveillance, causes anxiety, discomfort, and physical risks through repeated testing and monitoring, and often leads to active treatment anyway [7–9]. Any form of treatment, as well as the diagnostic workup, generates medical costs, often loss of income and inevitably opportunity costs [10]. Thirdly, overdiagnosis harms not only the individual patient but also society at large. Overdiagnosis drains healthcare resources from where they might be beneficial, and it obfuscates scientific data by inflating incidence rates and survival statistics through unnecessary diagnoses [11].

Screening for disease or risk factors is the most important driver of overdiagnosis. Still, overdiagnosis might also occur in clinical practice through incidental findings, broadening diagnostic criteria, or overselling common life experiences as diseases or medical entities. The ever-growing emphasis on early diagnosis and fast-track diagnostic packages in cancer diagnosis steadily increases the risk of overdiagnosis in clinical care [12]. The concept is also often confused with other forms of medical overuse (overtesting, overtreatment) or with diagnostic errors (false positives, misdiagnosis) [1]. The debate around the definition and the conceptual challenges of overdiagnosis is still ongoing and elucidates different opinions about the semantics of the definition and the operationalisation of the concept. Some authors believe that a key feature of overdiagnosis is that the condition would never have led to symptoms if left undetected, while others state that also diagnoses that ultimately have no utility or result

in more harm than benefit are overdiagnosed. Other points of debate are: should the notion of a potential benefit of overdiagnosis be allowed, and who should decide how harmful overdiagnosis is (the public or experts) [13–19].

Overdiagnosis occurs in several medical fields: metabolic and cardiovascular health, infectious diseases, and mental health, but it has only gained significant attention due to its notable effects in cancer screening [20]. However, the phenomenon is counter-intuitive and challenging to understand. Overdiagnosis happens in individuals, but it can only become visible at a population level.

Furthermore, researching and quantifying overdiagnosis in screening is difficult and necessitates long-term monitoring and reliable morbidity registers. It is also susceptible to bias and confounding factors, with outcomes heavily dependent on the researchers' fundamental assumptions [21–25].

Nevertheless, overdiagnosis is nowadays seen as a substantial harm of screening, which must be weighed against the potential benefits of screening, such as lower disease-specific mortality rates [26–28], reduced morbidity [29,30], or preservation of functionality [31]. Screening is inevitably taking a change for the future: the decision to screen for a particular condition is made now in the hope of benefiting in the future from an earlier diagnosis and treatment while avoiding possible negative consequences. At the time of screening, however, there is only uncertainty: about the individual risk of developing a detrimental disease, the chances of benefiting from an earlier diagnosis, and the risks of getting harmed due to screening. It is a difficult trade-off for individuals and the health professionals who guide them in these screening decisions. Eligible screening participants should be informed comprehensively and understandably about the benefits and harms of screening based on the best available evidence and encouraged to make an informed decision that aligns well with their preferences and values in health [32–36]. Many studies have examined laypeople's knowledge and perception of overdiagnosis. Almost all conclude that most people do not know about overdiagnosis and that the concept is difficult to understand [37–40]. Many participants in these studies felt ill-informed, expected public screening programs to be transparent about screening harms, and relied on their physicians to provide them with the necessary information [37,41,42].

However, findings from studies examining people's recollections of pre-screening discussions suggest that their physicians seldom provide information about screening harms [43–45]. A direct observation of German gynaecologists' counselling about breast cancer screening found that none of the gynaecologists mentioned the risk of overdiagnosis when the women inquired about the possible harms of having a mammogram [46]. Moreover, many health authorities organise mass screening programs, thus implicitly sending a message that screening is good for the public's health and well-being. Many of these official screening programs use framing effects to increase participation and do not communicate about the risk of overdiagnosis or do it only in veiled terms [47–51]. Finally, a recent systematic review of patient decision aids (PDA) for cancer screening found that one in five PDAs does not address overdiagnosis at all [52].

If people are to make an informed choice about screening, they need readily available information about this complex problem and time and space to reflect on it, free from steering or nudging. This puts a significant responsibility on health professionals (HPs) because they often initiate screening and should be the public's primary source of trusted information about possible screening benefits and harms. This assumes, however, that these HPs are aware of overdiagnosis and see it as a relevant harm for which they feel obliged to inform screening candidates.

With this systematic review, we aim to provide an overview of all available research about what health professionals know and think about overdiagnosis resulting from screening. The

research objective of this mixed-methods systematic review can be formulated in 4 research questions (RQs):

- Do HPs know about overdiagnosis resulting from screening? (Knowledge & awareness)

- How important do HPs consider overdiagnosis due to screening? (Perception of overdiagnosis)

- Does knowing about overdiagnosis affect HP's position towards offering screening? (Screening policy and practices)

- What do HPs think about providing information about overdiagnosis when offering screening? (Communication about overdiagnosis)

## Methods

This is a mixed-methods systematic review of qualitative and quantitative studies about the perspective of HPs on overdiagnosis resulting from screening. A thorough description of the methods of this systematic review has been published previously [53], and the protocol was registered on PROSPERO (CRD42021244513).

### Eligibility criteria

We included studies that directly questioned HPs on their awareness, knowledge, and perception of overdiagnosis resulting from screening. We broadly defined HPs as all professionals who, in the course of their professional activities, are involved in organising, offering, or performing screening. This encompasses public health (PH) officials and experts involved in population-based screening programs and individual healthcare providers (IHPs), e.g., clinicians who may talk about or offer screening to individual patients. Likewise, the concept of screening is also broadly defined as all forms of testing among asymptomatic people to find a disease early to improve the prognosis by early treatment. We included qualitative studies that offer insight into HPs' perceptions, considerations, and values, as well as quantitative studies to assess the magnitude of the findings. We excluded indirect observations of how HPs deal with overdiagnosis, research only examining laypeople's perspectives, and studies that focus on overdiagnosis unrelated to screening, such as overcalling common life experiences as mental disorders or broadening diagnostic criteria. We did not apply any language restriction or limitation in the time frame of publication.

### Search strategy, study selection, data extraction, and quality assessment

We conducted systematic searches in the following databases: MEDLINE (via PubMed), Embase, Web of Science, Scopus, CINAHL, and PsycArticles (via ProQuest). At the time of the conception of this systematic review, there was no MeSH-term for overdiagnosis, which was only introduced in 2022 [2]. The concept was categorised under the umbrella MeSH-term "overuse", which is appropriate but too broad. We, therefore, developed specific search strings to cover the concept of overdiagnosis, which are available as a supplement in our protocol article [53]. We also scanned reference and citation lists of the included articles and the first 50 'similar' articles proposed by the database search interfaces (forward and backward snowballing) and contacted experts in the field for additional publications. All databases were searched in June 2021. Searches were updated in May 2024, adding the new MeSH-term 'overdiagnosis'. Finally, we also hand-searched all abstracts of the annual conference on 'Preventing Overdiagnosis' (up to 2023).

Two researchers (VP and FS) independently screened all retrieved records for eligibility using a stepwise approach. All reasons for exclusion after full-text reading were recorded. Any

disagreement about inclusion was resolved through debate supervised by a senior researcher (SH). The entire selection process is reported in a PRISMA diagram.

We used a piloted data extraction form to collect all key bibliographical, contextual, and methodological data and a summary of the relevant findings. All manuscripts of qualitative studies and the qualitative components of mixed-methods studies were imported into NVivo software. The results from quantitative studies and the quantitative parts of mixed-methods studies were summarised in evidence tables. We performed a formal quality appraisal of all included publications: for the qualitative studies, we used the 'Framework for Assessing the Quality of Qualitative Research Evidence' [54]; for the quantitative studies, a 'Guide for appraising Survey Reports' [55], and for Delphi studies, the 'RAND Methodological Guidance for Conducting and Critically Appraising Delphi Panels' [56]. VP appraised all included articles, and a subset was independently evaluated by a senior researcher (AVH).

## Data analysis

In our protocol [53], we planned to analyse the findings with Critical Interpretive Synthesis (CIS) as the review method [57,58]. However, it was not possible to apply this approach for several reasons. First, our research question was already explicitly defined from the beginning, which left little room for the inductive development of the research question and related iterative literature search, as promoted by CIS. Second, we found only a few studies that provide rich data, most studies mention overdiagnosis only marginally, which leaves too scarce material for generating new overarching concepts. Third, the data from the quantitative studies were too limited, fragmented, and heterogeneous to integrate them in a meaningful synthesis with the qualitative data. We, therefore, analysed the qualitative and quantitative data separately and present them in two complementary syntheses [59]. The analysis of the qualitative data was inspired by the principles of Thematic Synthesis [60], but we also kept some features specific to CIS, such as allowing several study designs, combining qualitative and quantitative data, and using the perspective of a multidisciplinary research team during data analysis. VP and EL independently familiarised themselves with the included qualitative studies, and VP developed a first provisional coding tree with the four research questions (knowledge, perception, screening policy, and providing information) as the main 'branches' of the tree. To guarantee sufficient sensitivity for other concepts than those predefined by the research question, each article was coded with a double intention: to collect data directly related to the research question and to capture, in an inductive approach, additional concepts if these would emerge from the data. VP and EL revised the coding tree in consensus, and VP coded all relevant text fragments from the results sections of the studies. VP presented the first results of descriptive themes to the research team and several experts with different academic backgrounds (medicine, psychology, educational sciences, philosophy, and sociology) in several sessions, resulting in an iterative and reflective process, enriching and reorganising the descriptive themes, verifying that the themes keep reflecting the original data, and then presenting the results again to another group of experts. This process allowed us to stabilise the findings and develop an overarching analysis. VP reassessed the included publications again to check for coherence between the original data and the findings.

The quantitative data were imported into a spreadsheet, grouped per research question, and reported descriptively.

## Strength of evidence

We applied the GRADE-CERQual Framework criteria for assessing the strength of evidence of qualitative review findings. This indicates how reliable and generalisable the findings are

based on the methodological quality of the studies and the coherence, adequacy, and relevance of the data [61]. We could not find similar guidance to assess the strength of evidence from the questionnaire studies (beyond the methodological quality appraisal already described above).

## Results

We identified 9786 unique records and included 23 publications from 21 unique studies. Fig 1 presents a PRISMA flowchart of the search and exclusion process and an overview of the reasons for exclusion. Supplementary file S1 Table lists the reasons for exclusion after full-text reading. Reading all abstracts of the Preventing Overdiagnosis Conferences (2013 – 2023) did not yield any additional publication to include (see Table 1 – S1 Appendix).

### Description of the included publications

Eleven publications [62–72] report on nine qualitative studies: three publications are based on the same study [66–68], and one contains the qualitative part of a mixed-methods study [72]. Thirteen publications [72–84] report on quantitative studies, including the mixed-methods study mentioned above [72] and the quantitative results of a Delphi study [77] (see Table 1). There were 232 participants in the qualitative and 5925 in the quantitative studies. Only three studies had overdiagnosis as the primary research topic [64,67,69]. All other studies investigated general knowledge, beliefs, and practices related to certain screenings, and most of them mention overdiagnosis only marginally, except for one study that evaluated a new decision aid about breast cancer screening and spent a substantial part of the results section on the different opinions that arose among the participants around informing screening candidates about overdiagnosis [71].

All studies were conducted after 2000 and only in high-income countries. They all dealt with cancer screening, more specifically breast and prostate cancer screening. Two studies also included colorectal cancer [70,75], and one included cervical cancer [70]. We found no studies on other forms of population-based screening (e.g., neonatal screening) nor on other forms of opportunistic screening that individual healthcare providers can offer in their consultations (e.g., screening for diabetes, STD, melanoma, etc.).

All the studies about prostate cancer screening questioned HPs in their role as individual healthcare providers and inquired about their intentions and actions regarding offering

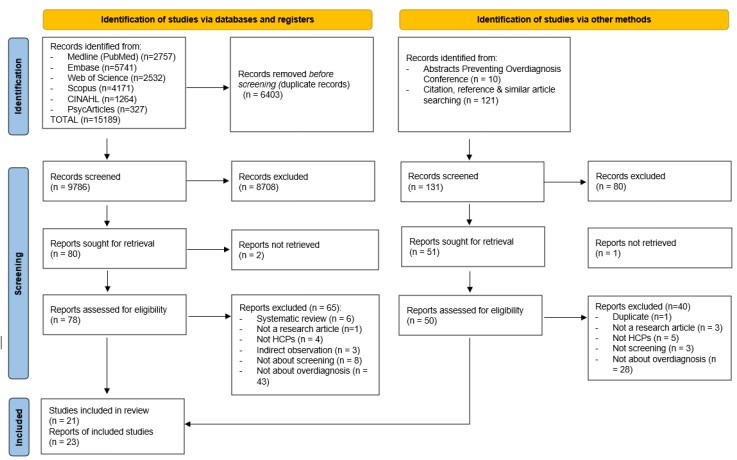

**Fig 1. PRISMA flow chart.**

**Table 1.** Overview of all included publications.

| Reference | Country | Screening topic | Overdiagnosis as the primary research topic? | Active screening program? | Guideline or professional recommendation regarding this screening? | Public Health (PH) or Individual Health Care (IHC) | Participants | Number of participants | Study method |
|---|---|---|---|---|---|---|---|---|---|
| Quantitative | | | | | | | | | |
| Akerman et al., 2018 [73] | Canada | Prostate cancer | No | No | Conflicting recommendations | IHC | General Practitioners (GPs) | 1880 | Questionnaire |
| Chan et al., 2003 [74] | USA | Prostate cancer | No | No | Yes | IHC | GPs, urologists, internists | 450 | Questionnaire |
| Elstad et al., 2015 [75] | USA | Prostate & colorectal cancer | No | Yes (colorectal cancer) | Yes | IHC | Primary care clinicians | 126 | Questionnaire |
| Goldenberg et al., 2017 [76] | Canada | Prostate cancer | No | No | Conflicting recommendations | IHC | Primary care physicians | 1190 | Questionnaire |
| Gunn et al., 2021 [77] | USA | Breast cancer | No | Yes | Yes | PH | Experts in breast screening, decision making or health literacy | 8 | Delphi |
| Kappen et al., 2019 [78] | Germany | Prostate cancer | No | No | Yes | IHC | GPs, urologists | 55 | Questionnaire |
| Kappen et al., 2020 [80] | The Netherlands | Prostate cancer | No | No | Yes | IHC | GPs | 88 | Questionnaire |
| Kappen et al., 2021 [79] | Germany | Prostate cancer | No | No | Yes | IHC | GPs, urologists | 528 | Questionnaire |
| Martinez et al., 2018 [81] | USA | Breast cancer | No | Yes | Yes | IHC | Primary care clinicians | 220 | Questionnaire |
| Petrova et al., 2017 [82] | UK | Cancer screening (hypothetical) | No | N/A | N/A | IHC | GPs | 151 | Experimental |
| Schoenberg et al., 2022 [83] | USA | Breast Cancer | No | Yes, but younger age group | Yes | IHC | Primary care physicians | 80 | Questionnaire |
| Shimada et al., 2017 [84] | Japan | Breast cancer | No | Not mentioned | Not mentioned | IHC | Nurses working in a breast screening clinic | 1710 | Questionnaire |
| Mixed methods | | | | | | | | | |
| Walters et al., 2010 [72] | UK | Breast cancer | No | Yes, but younger age group | Yes | PH + IHC | Breast cancer or geriatric experts | 139 (26) | Questionnaire + Interview |
| Qualitative | | | | | | | | | |
| Clements et al., 2007 [62] | UK | Prostate Cancer | No | No | Yes | IHC | GPs | 21 | Interview |
| Dois et al., 2021 [63] | Chili | Breast Cancer | No | Yes | Not mentioned | PH | Breast screening experts | 12 | Focus group |
| Gimenez, 2018 [64] | France | Breast Cancer | Yes | Yes | Not mentioned | IHC | GPs | 15 | Focus group |
| Malli, 2013 [65] | Austria | Prostate Cancer | No | No | Yes | IHC | GPs | 42 | Focus group + interview |
| Parker et al., 2015 [67] | Australia | Breast cancer | Yes | Yes | Not mentioned | PH | Breast screening experts | 33 | Interview |
| Parker et al., 2015 [66] | Australia | Breast Cancer | No | Yes | Not mentioned | PH | Breast screening experts | 33 | Interview |
| Parker et al., 2015 [68] | Australia | Breast cancer | No | Yes | Not mentioned | PH | Breast screening experts | 33 | Interview |

*(Continued)*

**Table 1.** (Continued)

| Reference | Country | Screening topic | Overdiagnosis as the primary research topic? | Active screening program? | Guideline or professional recommendation regarding this screening? | Public Health (PH) or Individual Health Care (IHC) | Participants | Number of partici-pants | Study method |
|---|---|---|---|---|---|---|---|---|---|
| Pickles et al., 2015 [69] | Australia | Prostate Cancer | Yes | No | Yes | IHC | GPs | 32 | Interview |
| Smith et al., 2022 [70] | Australia | Prostate, Colon, Breast, Cervical Cancer | No | Yes (except for prostate) | Yes | IHC | GPs | 28 | Interview |
| Toledo-Chávarri et al., 2017 [71] | Spain | Breast Cancer | No | Yes | Not mentioned | PH | Breast screening experts | 23 | Focus group |

opportunistic prostate cancer screening to their patients. These studies all took place in settings without population-based prostate screening programs and with different, sometimes contradicting, screening guidelines or professional recommendations. [62,65,69,70,73–76,78–80]. The studies about the other types of screening (breast, colorectal, and cervical) were held in settings where population-based screening programs were in place. Some studies examined IHPs in their role as implementers of these screening programs, while others took a public health perspective, asking professionals about their ideas on organising population-based screening and how to inform the public about it [63,66–68,71,72,77].

## Quality assessment

Table 2 presents the quality assessments of the quantitative studies. Most publications fail to provide information on questionnaire development and testing. Some studies report minimal or no efforts to minimise responder bias or provide too limited information on this item. Quality appraisal of the Delphi study by Gunn et al. [77] using the RAND methodological guidance [56] shows a high overall quality (see Fig 1 – S1 Appendix). Table 3 presents the quality appraisal of the qualitative studies. Four studies have good quality scores [66–71]. The reasons for the lower quality appraisals for the other studies are mainly lack of depth and richness, unclarity about underlying assumptions, theoretical frameworks and basis for evaluative appraisal, and unclear or biased sampling. No studies were excluded following the quality assessment.

## Findings

### Knowledge and awareness (RQ1)

None of the included studies intended to investigate whether HPs had an accurate *knowledge* of the concept of overdiagnosis. Only one qualitative study reports briefly on the participants' knowledge and notes that the concept of overdiagnosis (in breast cancer screening) seemed poorly understood and confused with false positives [64]. Five of the 13 included quantitative studies report on HPs' *awareness* of overdiagnosis [72,73,75,81,84]. However, there is significant heterogeneity in how the survey questions were formulated, and some questions did not purely assess HPs' awareness of overdiagnosis but combined it with the concept of overtreatment (see Table 1 in S2 Table).

When the investigators actively address overdiagnosis in their questionnaires, awareness rates range from 57% to 92% [72,73,81,84]. The one study that did not prompt respondents

**Table 2. Quality appraisal of the quantitative studies (Guide for appraising Survey Reports[47]).**

| | Was a clear research question posed? | Was the target population defined, and was the sample representative of the population? | Was a systematic approach used to develop the questionnaire? | Was the questionnaire tested? | Were questionnaires administered in a manner that limited both response and nonresponse bias? | Was the response rate reported, and were strategies used to optimize the response rate? | Were the results clearly and transparently reported? |
|---|---|---|---|---|---|---|---|
| Akerman et al., 2018 [73] | + | + | – | – | + | + | +/– |
| Chan et al., 2003 [74] | + | + | +/– | +/– | + | + | +/– |
| Elstad et al., 2015 [75] | + | + | +/– | +/– | +/– | + | + |
| Goldenberg et al., 2017 [76] | + | + | + | +/– | + | +/– | +/– |
| Kappen et al., 2019 [78] | + | + | – | +/– | + | +/– | + |
| Kappen et al., 2021 [79] | + | + | – | +/– | – | +/– | + |
| Kappen et al., 2020 [80] | + | + | – | +/– | + | +/– | + |
| Martinez et al., 2018 [81] | + | + | – | – | +/– | +/– | + |
| Petrova et al., 2017 [82] | + | + | +/– | +/– | + | + | + |
| Schoenberg et al., 2022 [83] | + | + | – | – | + | +/– | + |
| Shimada et al., 2017 [84] | + | + | +/– | – | +/– | +/– | + |
| Walters et al., 2010 [72] | + | + | +/– | – | +/– | +/– | + |

"– " = no or not reported, "+" = yes, "+/– " = partially yes, partially no or not reported

on overdiagnosis but instead asked them to list all adverse effects of screening found that only 28% of HPs spontaneously mention overdiagnosis as a harm in prostate cancer screening, and 8% do so for colorectal cancer screening. There is no further information about what the participants understood under the term overdiagnosis [75].

Similarly, in most qualitative studies, the investigators actively brought overdiagnosis up as a topic. However, when not prompted by the researchers, participants seldom mention overdiagnosis spontaneously [63,65,70]. Nevertheless, many citations from different qualitative studies suggest that at least some participants have accurate knowledge about overdiagnosis [62,63,68–72].

> "We are looking for smaller and smaller lesions, we are picking up little bits of micro calcification, we are bringing women back for repeat biopsies... for the detection of smaller and smaller lesions that may never even have been invasive cancer. And even if they had been invasive cancer, they might never have threatened their lives. (Radiologist)"[72]

In one study, some participants contest the concept and state that overdiagnosis is a phenomenon in which the diagnostic biopsy already removed the cancer entirely, after which no cancer would remain, leaving only the mistaken conclusion that it was an unnecessary diagnosis [71].

**Table 3. Quality appraisal of the qualitative studies (Framework for Assessing the Quality of Qualitative Research Evidence[46]).**

| | | Clements et al., 2007 [62] | Dois et al., 2021 [63] | Gimenez, 2018 [64] | Malli, 2013 [65] | Parker et al., 2015 [66–68] | Pickles et al., 2015 [69] | Smith et al., 2022 [70] | Toledo-Chávarri et al., 2017 [71] | Walters et al., 2010 [72] |
|---|---|---|---|---|---|---|---|---|---|---|
| Findings | How credible are the findings? | + | + | +/− | +/− | + | + | + | + | + |
| | How has knowledge/ understanding been extended by the research? | + | + | +/− | +/− | + | + | + | + | + |
| | How well does the evaluation address its original aims and purpose? | + | + | + | + | + | + | + | + | + |
| | Scope for drawing wider inference – how well is this explained? | + | +/− | +/− | +/− | + | + | + | + | +/− |
| | How clear is the basis of evaluative appraisal? | +/− | − | +/− | − | +/− | +/− | +/− | +/− | − |
| Design | How defensible is the research design | + | + | +/− | + | + | + | + | + | +/− |
| Sample | How well defended is the sample design/ target selection of cases/documents? | +/− | +/− | +/− | + | + | + | + | + | + |
| | Sample composition/case inclusion – how well is the eventual coverage described? | +/− | +/− | +/− | +/− | + | +/− | +/− | + | + |
| Data collection analysis | How well was the data collection carried out? | +/− | + | + | + | + | + | + | + | +/− |
| | How well has the approach to, and formulation of, the analysis been conveyed? | + | + | +/− | + | + | + | + | + | +/− |
| | Contexts of data sources – how well are they retained and portrayed? | + | + | +/− | + | + | + | + | + | + |
| | How well has diversity of perspective and content been explored? | + | +/− | +/− | +/− | + | + | +/− | + | +/− |
| | How well has detail, depth and complexity (i.e., richness) of the data been conveyed? | + | +/− | − | + | + | + | + | + | +/− |
| Reporting | How clear are the links between data, interpretation and conclusions – i.e., how well can the route to any conclusions be seen? | + | +/− | − | + | + | + | + | + | +/− |
| | How clear and coherent is the reporting? | + | + | +/− | + | + | + | + | + | + |
| Reflexivity & neutrality | How clear are the assumptions/theoretical perspectives/values that have shaped the form and output of the evaluation? | +/− | − | − | +/− | + | +/− | +/− | + | − |
| Ethics | What evidence is there of attention to ethical issues? | − | + | +/− | +/− | + | +/− | +/− | +/− | − |
| Auditability | How adequately has the research process been documented? | +/− | +/− | +/− | +/− | + | +/− | + | +/− | +/− |

"– " = not good or not reported, "+" = good, "+/− " = partially good, partially not good or not reported

### Overarching analysis: an inextricable interplay between the perception of overdiagnosis and overall beliefs about screening

The qualitative findings show that HPs familiar with the phenomenon of overdiagnosis vary substantially in the extent to which they consider it a significant problem, even when they use a similar definition for the concept. HPs' perceptions of overdiagnosis (RQ2) appear inextricably intertwined with HPs' broader beliefs about the benefits and harms of screening and whether screening should be offered to the public. HPs who pay much attention to the harmful consequences of overdiagnosis also tend to be more sceptical about the benefits of screening and are more reluctant to offer it. Conversely, HPs who strongly believe in the

life-saving potential of screening and advocate that as many people as possible should be screened seem to see overdiagnosis as a minor issue with little or no health consequences. These beliefs are on a sliding scale, and the correlation between perceptions of overdiagnosis and general beliefs about screening cannot be considered a one-to-one match. These ideas also correlate with HPs' position towards offering screening (see RQ3) and their willingness to discuss the risk of overdiagnosis (see RQ4). Fig 2 illustrates this overarching analysis in a schematic overview of how the perception of overdiagnosis is connected to overall screening beliefs, preferred screening policies, and willingness to communicate about overdiagnosis.

## Perception of overdiagnosis (RQ2)

Only one quantitative study provides limited information about how harmful HPs perceive overdiagnosis. Elstad et al. asked participants to list all the disadvantages of screening they could think of and to indicate how large they believed the harm would be, ranging from 1 (almost no harm) to 4 (very harmful). The mean harm from overdiagnosis is perceived as 3.48 for prostate screening with PSA and 2.67 for colorectal screening with colonoscopy [75].

Four qualitative studies (6 publications) give insight into HPs' diverging perceptions of overdiagnosis [66–69,71,72]. The differences depend mainly on how HPs perceive the potentially harmful consequences of overdiagnosis and can be described in five categories: (1) *overdiagnosis is a devastating harm, (2) overdiagnosis is a minor harm, (3) overtreatment is the problem, not overdiagnosis, (4) overdiagnosis is a normal consequence of screening, (5) overdiagnosis is not a harm, may even be beneficial.*

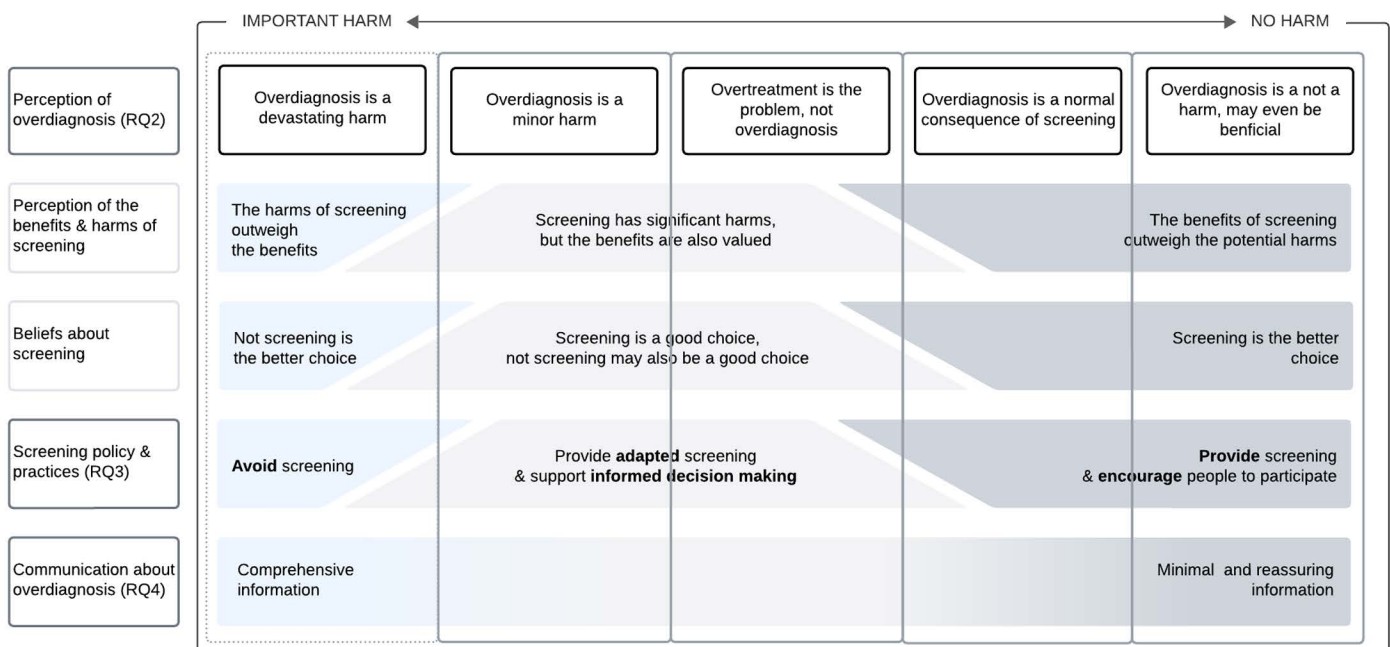

**Fig 2. Different perceptions of overdiagnosis and related ideas about the benefits and harms of screening, preferred screening policies, and communication strategies.**

**Perception 1: "Overdiagnosis is a devastating harm".** HPs who see overdiagnosis as the main risk of screening believe it is a significant threat to a person's health and well-being because of the harmful mental consequences of a cancer diagnosis as well as from the painful and sometimes mutilating treatment. They feel the overdiagnosis rate is too high, especially compared to the low number of lives saved. Some state that the risk of overdiagnosis outweighs any potential benefit of screening, while others believe that both facts are true: screening can save lives, and screening can cause significant damage [66,67,69,72].

*These GPs believed the harms of overdiagnosis were too great to justify testing, "Even though we—in the long term you might save someone's life, if you do an awful lot of harm along the way, it's just not worth it" [69]*

*Harm quantity was described in terms of the high number of overdiagnosed cases compared to the number of lives saved by screening. Harm quality was discussed by highlighting the serious negative impact from each case of overdiagnosis, including both the psychological impact of a breast cancer diagnosis on a woman and her female relatives (for whom it has perceived risk implications), and the short and long term impact of unnecessary treatment on lifestyle and physical health [67]*

**Perception 2: "Overdiagnosis is a minor problem or a necessary evil".** Other HPs perceive the number of overdiagnosed cancers as relatively low compared to all the cancers that can be detected through screening and believe treatment options for overdiagnosed cancers are not that invasive [67].
Some HPs describe overdiagnosis as a 'necessary evil': they still acknowledge some harm but see it as an unavoidable downside of screening that must be accepted to obtain its benefits [69,72].

*[These GPs] thought testing was necessary because there was a possibility it might prevent a man's death. Overdiagnosis was perceived as (1) a natural consequence of PSA testing; (2) better than dying and (3) a justifiable source of harm (harms being a regrettable but necessary price of 'cure'). [69]*

**Perception 3: "Overtreatment is the problem, not overdiagnosis".** Some HPs mainly emphasise the harmful consequences of overly aggressive treatment choices. They ignore any adverse impact of the cancer diagnosis itself [67–69].

*Some of these GPs thought decisions about postdiagnosis management (e.g., active surveillance) could limit the harms of potential overdiagnosis. This allowed them to define testing without invasive procedures as inconsequential: "it's not terribly onerous to have a blood test every six months"[69]*

**Perception 4: "Overdiagnosis is a normal consequence of screening".** These HPs disregard any notion of harm in overdiagnosis. Instead, they see it as the intended outcome of a screening program: detecting more cancers to improve the prognosis for a larger population [67,68,71].

*Some experts added to this by asserting that overdiagnosis was not a harm, rather that the diagnosis of small cancers was exactly what the screening program was intended to do in order to reduce breast cancer mortality and morbidity [68].*

**Perception 5: "Overdiagnosis is not a harm; it may even be beneficial".** These HPs argue that overdiagnosis does not correspond to patients' definitions of screening harm but is rather a fictitious academic construct that holds no connection with people's reality [68].

> *Some stated that the concept of overdiagnosis being a harm was based on opinion, rather than fact and therefore did not count as information. "Harm is a term that's been developed by academics, along academic lines… [Overdiagnosis is not] women's definition of harm" [68]*

For some HPs, overdiagnosis may even be beneficial because any information about a person's health is considered intrinsically valuable. Moreover, the detection of an overdiagnosed cancer may indicate an increased susceptibility to cancer for which one can then protect oneself [67,69].

> *In addition to the lack of harm, the frame highlighted possible benefits from overdiagnosis. Although, by definition, an overdiagnosed cancer will not itself threaten a woman's life, experts suggested that as the woman would be at increased risk of a second breast cancer she would benefit from being identified and treated with tamoxifen. [67]*

## Screening policy and practices (RQ3)

No quantitative studies investigated whether awareness of overdiagnosis affects HP's position towards offering screening. In the qualitative findings, three distinct screening approaches emerge from the data: (1) *avoiding screening*, (2) *providing (adapted) screening with explicit efforts to support informed decision-making*, and (3) *offering screening and encouraging people to participate*. HPs' preference for a specific screening policy is linked to their perception of overdiagnosis and the extent to which they value the benefits of screening (see Fig 2). Depending on the HP's role, these screening approaches may be differently operationalised. HPs who speak from a public health perspective express their ideas about how population-based screening programs should be organised, while IHPs explain how they act personally in interactions with their patients. Table 4 presents some citations illustrating the three screening approaches from the PH and the IHP perspective.

**Avoid screening.** HPs who see overdiagnosis as a vital harm that prevails largely over the potential benefits of screening prefer to avoid screening. For public health professionals, this could mean rolling back existing population-based screening programs or stopping the public promotion of these programs. They strongly believe that public health programs should not cause harm to the public [66,67,69]. For individual healthcare providers, this means not proactively offering screening themselves or actively discouraging their patients from getting screened. Their primary objective is not to inflict harm on their patients. However, they are well aware that avoiding screening may lead to missed opportunities to save lives, leading to a substantial mental burden for these HPs. They also try to ignore their medicolegal concerns and have to cope with the fear of being held responsible for missing a cancer diagnosis [62,69].

**Provide adapted screening and support informed choice.** HPs who value the benefits of screening while being aware of overdiagnosis prefer to offer screening but insist that efforts should be made to support citizens in making informed screening decisions [64,67–69,71]. They believe screening candidates should have a predominant voice in these decisions because harms and benefits can be equally relevant but differently appreciated depending on individual values. HPs favouring this screening approach believe that people have the right to be fully informed and want to avoid a paternalistic attitude. This contrasts with the two other approaches in which HPs (tacitly) see it as their responsibility as experts to decide about screening and advise people on what is best for their health. Other HPs try to mitigate the

**Table 4. Citations illustrating different approaches to screening.**

| | *Avoid screening* | *Provide (adapted) screening and support informed choice* | *Offer screening and encourage people to participate* |
|---|---|---|---|
| **Public health** | *Experts who were more sceptical about the benefits accruing from breast screening preferred a more extreme solution: reducing overdiagnosis by decreasing overall breast screening participation [... … …] however] they assumed that cessation of public funding for the program was politically unlikely [67]* <br> *This framing of overdiagnosis as a serious problem was grounded in a strong commitment to avoiding harm in any public health program [67]* | *"This is not straightforwardly a good thing. There are some downsides and while we don't necessarily think the downsides are such that you shouldn't be doing it, at the very least, we should be telling women about this so that they can make an informed decision." [68]* <br> *On an ethical level, they posited that it is imperative to inform people about the available evidence because these data change the balance between the benefits and harms [71].* <br> *Experts who were enthusiastic about the potential benefits of screening suggested reducing overdiagnosis through a targeted, personalised screening program, matching recommended screening frequency to breast cancer risk as determined by factors such as breast density. This would enable the population to simultaneously retain benefits of screening and reduce harms. [67]* | *"Breast screening commentators should give priority to delivering health benefits (saving lives) The frame delivers a choice between life and overdiagnosis: "saving a life is more important than the harm that's caused in damaging normal breasts." [67]* |
| **Individual healthcare** | *GPs [… ….] preferred not to conduct PSA testing. Their primary justification was preventing harms caused by overdiagnosis. However, while they would try to talk patients out of having the test, they would never refuse a PSA test. These GPs also recognised that PSA testing has saved lives; "we know that happens. The problem is, it just doesn't happen often enough to balance out…all the damage that we do" [69]* | *Some GPs wanted to avoid a paternalistic attitude. They preferred to provide full information, including on risks without fear of reducing screening participation [64].* <br> *Some of these GPs thought decisions about postdiagnosis management (e.g., active surveillance) could limit the harms of potential overdiagnosis. This allowed them to define testing without invasive procedures as inconsequential: "it's not terribly onerous to have a blood test every six months" [69]* | *These GPs focused on cancer as life-threatening, and prostate cancer as a terrible death. They saw preventing death as the primary duty of the GP. This heightened their responsibility to do anything that may diagnose cancer early: "Because if you don't overdiagnose, the alternative is to underdiagnose") [69]* |

harms of overdiagnosis by proposing adaptations to the existing screening program, such as reducing the subsequent treatment harms or moving to personalized screening of only high-risk individuals [64,67,69].

**Offer screening and encourage people to participate.** HPs who do not perceive overdiagnosis as a significant harm or for whom the benefits of screening are paramount do not consider changing screening policies, prefer to continue offering screening and encourage people to participate. Their main objective is to save lives [66,67,69,71].

## Communication about overdiagnosis (RQ4)

Eight quantitative studies gauged HPs' ideas about informing screening candidates about overdiagnosis (see Table 2 in S2 Table). Six studies were held among individual healthcare providers, of whom the majority indicated personally discussing overdiagnosis before screening (69% to 83% of participating HPs) [76,78–80,82,83]. In two other studies, researchers asked HPs if they believed that screening candidates should know about overdiagnosis before getting screened (without necessarily having to discuss it themselves, e.g., the information should be part of a patient decision aid) and found similar results (over 85% of HPs believed people should be informed about overdiagnosis)[74,77].

In contrast, the qualitative studies show that HPs have varied approaches to addressing overdiagnosis, with significant differences in content, amount of detail, and connotation of harm. Their preferences for providing information about overdiagnosis align with their general idea of the net benefit of the proposed screening, their perception of overdiagnosis, and what "taking good care of their patients or the public" means to them as professionals (see Fig 2).

**Comprehensive information about overdiagnosis.** On one side of the spectrum, HPs believe it is their duty to thoroughly inform people about the risk of overdiagnosis [63,64,66–71]. The information must be comprehensive and easy to understand. However, they recognize that it is a complex topic, even for professionals, one that requires repeated

and careful communication before it can be fully understood. Some HPs deliberately highlight overdiagnosis as a potential harm and use this as a strategy to discourage people from screening to protect them from potential harm [68–70].

> *Some GPs were strongly oriented to avoiding overdiagnosis, and so tried to test as little as possible. This group of GPs emphasised the harms of PSA testing (including overdiagnosis) when advising their patients; and said many patients chose not to be tested following discussion. These GPs, who fully explain overdiagnosis, described themselves as "taking the risk of doing the hard work, hard yards" [69]*

Other HPs want to provide comprehensive information about overdiagnosis to support an individual's autonomy to make an informed choice about screening options, given the tight balance between the benefits and harms of screening. They see it as their moral duty to be transparent and honest about benefits and harms [63,64,66–71].

> *They engaged patients in detailed discussions about potential harms and resisted medicolegal fears. "If I did that…I think I would be a more paternalistic doctor who ordered a lot more tests. And I don't see that would be good medicine. I think it would do more harm to more people for practicing defensively like that" [69]*

> *Contrary to this position, a smaller group of experts advocated full information about both benefits and harms of breast screening. They particularly wanted consumers to be provided with understandable data about overdiagnosis, including readily comparable information on chances of mortality benefit versus overdiagnosi.[68].*

**Limited and reassuring information about overdiagnosis.** On the other end of the spectrum, HPs prefer to give no or only limited and reassuring information about overdiagnosis, preferably after people have already engaged in getting screened. Some HPs are worried that comprehensive information would lead to confusion and anxiety among the public, especially because the concept is difficult to explain and evidence translation from population statistics is complex and uncertain [67,69,71].

> *They suggested overdiagnosis information should be presented briefly along the lines of, "some of the things that we are going to be treating you for may not progress." (Expert #33, clinician and provider) These experts proposed that further information could be made available for those who wanted it. [68]*

> *They were in favour of informed participation and SDM, although some specified that their support was limited to situations that did not generate fear or confusion. [71]*

Others are reluctant to discuss overdiagnosis out of concern that it would scare people away from participating in screening. These HPs' primary objective is to save lives, and they strongly believe that anything that could hamper screening participation should be avoided [66–68,71].

> *Overdiagnosis and overtreatment risks can't be shared with women because participation would decrease. Imagine if you inform that it should not be undertaken… then no one would come (doctor in a cancer research unit) [71].*

**Overdiagnosis not mentioned in pre-screening discussions.** A few HPs believe information about overdiagnosis is not relevant in the pre-screening phase. For these HPs,

it should only be discussed when considering treatment options in case the screening would have led to a diagnosis [62,69].

> *"I feel I'm in second line for that (discussion of range of prostate cancers) because if they go ahead and have the biopsy, say they have the PSA test and it's positive then you have to, you are obliged to refer... the urologists are obliged then to investigate further.....so it's very much their stance isn't it?"* [62]

Finally, many HPs seem not to talk about overdiagnosis, not necessarily because they deliberately choose to avoid the topic, but rather because overdiagnosis is not on the list of topics they usually address in pre-screening discussions [62,63,65,69].

> *The majority of GPs in this group did not engage with considering the implications of underdiagnosis or overdiagnosis and what that meant for their patients. […] GPs do not talk about overdiagnosis or underdiagnosis [69].*

Supplementary file S3 Table provides an overview of the data extraction by included publication, and S4 Table shows all extracted data of the qualitative studies, organised by review finding.

## Strength of evidence

Because of the paucity of studies specifically investigating HPs' perspective on overdiagnosis, all findings are graded with low to moderate certainty. The concerns are mainly related to adequacy (quantity and richness of the data) and relevance (the extent to which the data are sufficient to answer the review question). Supplementary file S5 Table shows a detailed assessment of the strength of evidence using the GRADE CERQual approach.

## Discussion

We found no studies directly examining HPs' knowledge about overdiagnosis. However, indirect evidence from the qualitative studies suggests that at least some HPs have accurate knowledge about overdiagnosis, while others appear to conflate it with false positives or other screening phenomena. HPs vary substantially in the extent to which they consider overdiagnosis a relevant problem, from viewing it as a significant harm to considering it a non-issue. Their perception seems connected with their overall beliefs about screening benefits and whether screening should be offered to the public. Questionnaire studies indicate that most HPs believe informing screening candidates about overdiagnosis is of importance. The qualitative studies, however, reveal a significant variation in how HPs are willing to communicate about overdiagnosis, from very comprehensive information to no information at all.

A major finding from this review is the paucity of evidence on what HPs know and think about this significant risk of screening. Only three of the included publications have overdiagnosis as the primary research topic and only 16 of the 38 assessed articles on HPs' view on the benefits and harms of screening mention overdiagnosis as a potential harm, and often only marginally (see S1 Table). This contrasts with the numerous studies investigating laypeople's knowledge [37–42]. This could indicate that within the research community, there may be some silent assumptions about HPs' perspectives on overdiagnosis or perhaps even a reluctance to question their knowledge. Another explanation could be that overdiagnosis is a blind spot for researchers, too. This hypothesis was raised earlier when reviews of original cancer screening trials and Cochrane reviews[85,86] found that in only a minority of the publications, overdiagnosis was addressed as a possible harm.

Our findings also show that HPs hold a broad range of attitudes towards overdiagnosis. However, it was often hard to separate their perspective on overdiagnosis from ideas about other, often more tangible harms of screening, such as false positives and direct harm of diagnostic follow-up. Moreover, their beliefs about the harms of screening seem deeply connected with their general idea about the value of screening. It is unclear whether there is a causal relationship here, and if so, in which direction. One of the included studies by Elstad et al. [75] suggests that HPs' overall 'gist' of screening is primarily formed by their perception of the benefits of screening, not so much by their awareness of the harms. Similar conclusions can be drawn from an experimental study by Petrova et al. [87] that found that physicians in training's recommendations for (a hypothetical) screening were primarily based on how they perceived the benefits of this screening, not on the potential risks.

In addition to this primarily cognitive explanation of the observed differences in attitude towards screening and overdiagnosis, other intrapersonal or societal factors may be equally or even more pertinent. Similar research among laypeople has already shown that accurate knowledge likely plays only a minor role in attitude towards screening [88]. The dominant societal discourse is that screening saves lives, and good citizens are expected to take care of their health by participating in regular screening. When people are presented with facts about screening that go against their preconceptions, a perception gap arises between the information provided and their interpretation [89]. People have internalized the societal screening messages, and questioning these beliefs may cause disbelief and strong affective resistance [90–92]. HPs are subject to the same dominant discourse and normative messages. We hypothesize that HPs, as fellow citizens in this society, react similarly to their lay counterparts and that non-cognitive intrapersonal factors, such as emotions, pre-existing ideas, and normative beliefs, could explain part of the variation we found in HPs' attitudes towards overdiagnosis [93–95]. A third explanation for the observed differences may be sought in HPs' professional roles and responsibilities. HPs' professional role comes with more knowledge, more socialising in the medical screening discourse, and interpersonal challenges, such as dealing with perceived patient expectations or taking a position among colleagues. HPs also carry the responsibility to guide or even decide for their patients, which may affect their position towards screening beyond their personal beliefs. Former experimental research found that doctors are more risk-averse for their patients than for themselves and even make more cautious decisions for their patients than patients would make for themselves [96].

We found that HPs' tailor their communication about overdiagnosis to what they try to achieve regarding screening. Informing screening candidates about overdiagnosis (or not) appears to have a broader function than merely providing information. HPs also seem to use their communication about overdiagnosis strategically to steer people's screening behaviour following their own professional standards and ethical principles. These findings align with other studies examining how HPs communicate about screening and how this relates to their screening practices and underlying beliefs and intentions. Driedger et al. [97] observed that family physicians who frequently ordered PSA tests engage much less in shared decision-making than their colleagues with a median ordering profile. Other studies also found that doctors who are critical of the evidence base that underpins screening are more likely to practice shared decision-making and use decision aids [98,99]. Pickles et al. [100] describe four overarching approaches to how Australian general practitioners (GPs) communicate about PSA screening. These approaches translate GPs' goals regarding screening, the level of understanding they try to achieve in their patients, and the type of information they want to convey.

Finally, we noticed critical differences in results depending on the study methodology and whether the researchers proactively prompted the issue of overdiagnosis. The quantitative studies, for example, find a high willingness among the participants to inform screening

candidates about overdiagnosis. This is in sharp contrast with, e.g., the qualitative study by Malli et al. [65] where the researchers state that none of the participants spontaneously mentioned discussing overdiagnosis in pre-screening counseling. Several explanations exist for these diametrical findings, such as ambiguous or even leading questions, socially desirable responses, availability bias, and confusing overdiagnosis with other harms of screening [1].

Our work has several strengths and limitations. We conducted a thorough search in different databases without any language restriction and combined qualitative and quantitative research. We did not limit our search to publications about overdiagnosis specifically but looked broadly at all studies that examined HPs' ideas about screening in general, assuming that questions about overdiagnosis may be a part of a broader inquiry after HPs' perspective on screening.

However, we did not look for grey literature and only found studies conducted in high-income countries.

These review findings need confirmation and further exploration of the reasons for the wide variation in attitude towards overdiagnosis we identified. The potential interplay between the societal screening discourse, HPs' intrapersonal factors, interpersonal interactions, and their professional behaviour may be underestimated and deserves research attention. The observed variation in beliefs about overdiagnosis and screening also calls for HPs to be aware of their potential knowledge gaps, to recognize that their position toward screening may not be self-evident, could even differ substantially from the position of their colleagues, and may affect their screening practices and the quality of the pre-screening information they provide.

## Conclusion

There is almost no data about how well health professionals (HPs) know overdiagnosis in screening, but when HPs do seem to know it, they attribute largely different levels of harm to it. This may be mediated by how HPs weigh the balance between the benefits and harms of screening and whether they believe that screening should be offered to the public. These beliefs may also guide them in how much detail they provide about overdiagnosis in pre-screening information. These review findings suggest that the public's right to reliable and evidence-based screening information is not always guaranteed but depends considerably on the level of knowledge and beliefs of the health professionals involved.

## Supporting information

**S1 Table.** Eligibility of publications.
(DOCX)

**S1 Appendix.** Selection QA.
(DOCX)

**S2 Table.** QN results.
(DOCX)

**S3 Table.** Data extraction by publication.
(DOCX)

**S4 Table.** Data extraction by review finding.
(DOCX)

**S5 Table.** GRADE CERQual.
(DOCX)

## Acknowledgements

We thank Dr. Nele S. Pauwels (Knowledge Centre for Health Ghent, Belgium) for her guidance in the development of the search strategy, and Prof. Dr. Heidi Mertes (Dpt of Philosophy and Moral Sciences, Ghent University), Prof. Dr. Pauline Boeckstaens, Prof. Dr. Peter Decat, Prof. Dr. Peter Pype (Dpt of Public Health and Primary Care, Ghent University), Prof. Dr. Piet Bracke (Dpt of Sociology, Ghent University), and Dr. Stijn Suijs (The Hague University of Applied Sciences) for their time and wisdom in interpreting and reflecting on the results.

## Author contributions

**Conceptualization:** Veerle Piessens, Ann Van den Bruel, An Piessens, An De Sutter.

**Data curation:** Veerle Piessens.

**Formal analysis:** Veerle Piessens, Emelien Lauwerier, Florian Stul.

**Funding acquisition:** An De Sutter.

**Investigation:** Veerle Piessens, Ann Van Hecke, Emelien Lauwerier, Florian Stul, Stefan Heytens.

**Methodology:** Veerle Piessens, Ann Van den Bruel, Ann Van Hecke, Stefan Heytens.

**Project administration:** Veerle Piessens.

**Supervision:** Ann Van den Bruel, Stefan Heytens.

**Validation:** Veerle Piessens.

**Visualization:** Veerle Piessens.

**Writing – original draft:** Veerle Piessens.

**Writing – review & editing:** Veerle Piessens, Ann Van den Bruel, An Piessens, Ann Van Hecke, John Brandt Brodersen, Emelien Lauwerier, Florian Stul, An De Sutter, Stefan Heytens.

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
