## [Decision Letter · Decision Letter 0]

21 Aug 2024

PONE-D-24-22152Do health professionals know about overdiagnosis in screening, and how are they dealing with it? A mixed-methods systematic scoping review.PLOS ONE

Dear Dr. Piessens,

Thank you for submitting your manuscript to PLOS ONE. After careful consideration, we feel that it has merit but does not fully meet PLOS ONE’s publication criteria as it currently stands. Therefore, we invite you to submit a revised version of the manuscript that addresses the points raised during the review process.

We look forward to receiving your revised manuscript.

Kind regards,

Pascal A. T. Baltzer, M.D.

Academic Editor

PLOS ONE

Journal Requirements:

https://bmjopen.bmj.com/content/12/10/e054267.full

In your revision ensure you cite all your sources (including your own works), and quote or rephrase any duplicated text outside the methods section. Further consideration is dependent on these concerns being addressed.

Reviewers' comments:

Reviewer's Responses to Questions

**Comments to the Author**

1. Is the manuscript technically sound, and do the data support the conclusions?

Reviewer #1: Yes

2. Has the statistical analysis been performed appropriately and rigorously?

Reviewer #1: Yes

3. Have the authors made all data underlying the findings in their manuscript fully available?

Reviewer #1: Yes

4. Is the manuscript presented in an intelligible fashion and written in standard English?

Reviewer #1: Yes

5. Review Comments to the Author

Reviewer #1: Dear authors

Thank you for the opportunity to read this ambitious manuscript, which involved searching for all research about health professionals’ (HPs) knowledge and perception of overdiagnosis, and analysing the relationship between this knowledge and perception and HPs intentions and actions with respect to screening.

Overall, this is a well-conducted review that makes an important contribution to the literature.

I offer the following comments to improve the manuscript.

1. Fundamental concepts and scope

There appears to be a disjunct between the introduction and the rest of the manuscript; there are also some conceptual inconsistencies within the introduction.

First, the introduction presents one conception of overdiagnosis when there are a range of definitions or conceptualisations which have continued to evolve in recent years. Given that this paper is fundamentally about what HPs take overdiagnosis to be, suggest it is wise to start by recognising that the concept and its definition is contested even among experts (whether or not all experts recognise this contestation!).

Further, the introduction states that overdiagnosed people “cannot benefit from this early diagnosis” (p.3 L57) and that treating a person with an overdiagnosed condition “cannot provide any benefit” p. 3 L65), but also states that the harms caused by overdiagnosis “must be weighed against the benefits of screening” (p.4 L82). The manuscript itself presents nuanced accounts of the complex benefit/harm balance involved in understanding and identifying overdiagnosis. Arguably this nuance is central to the main summary figure, Fig2. I would suggest this more nuanced approach is more consistent with the reality of what clinicians (and their patients) face and should be reflected in the introduction.

Finally, the manuscript uses the word screening without explaining how the study is conceptualising screening, or the imagined relationship between population screening and opportunistic testing. Your primary data seems to contain the views of both clinicians (who may be offering opportunistic tests to individuals, or may be informing individuals about population based screening programs) and administrators and others with an interest in population based programs. Suggest this screening-opportunistic testing distinction should be addressed directly, and you should make sure that when you report findings you are clear whether you are talking about a clinician (e.g. a GP) or an administrator (e.g. a program manager). This is especially pertinent to your third RQ which refers to ‘offering’ screening, but it is not completely clear what constitutes an offer in the context of the review and these different contexts.

Note that addressing these comments is not just about editing the introduction, but also about reflecting on how the concepts were operationalised in the the course of undertaking the review, and being sure that these were internally consistent and are consistent with the finished Introduction.

2. Methods

As noted, the review is generally very strong.

There are reasons given for abandoning CIS in the discussion, but I wonder whether this information may be better in the methods? As there appears to be a significant amount of inductive analysis involved in the review, I was not 100% clear why CIS was abandoned. I was also not clear whether you would describe the methodology you followed as bespoke, or whether you found an alternate methodology? Some justification for why you did things the way you did them, if it was a bespoke method, would be helpful.

Can you say more about why you didn’t assess quality of the quantitative studies?

3. Findings

The knowledge/awareness results seemed thin, although this is arguably compensated by later sections.

The most substantive section is the ‘perception of overdiagnosis’ section. This section appears to rely heavily on a small number of the studies found based on the quotations used. If this is the case, suggest addressing this more directly. The important ‘perceptions of overdiagnosis’ section also appears to rely heavily on quotes rather than presenting very much new analysis from the authors of this review. Can you provide any further analytic depth that draws out new insights by comparing & further synthesising the primary studies?

The quantitative and qualitative findings seem quite disconnected in the results section – is it possible to connect them any further?

Figure 2 is useful and interesting and will be an important contribution of the study.

The discussion is a strength of the MS and raises some important and useful implications.

6. PLOS authors have the option to publish the peer review history of their article (what does this mean? ). If published, this will include your full peer review and any attached files.

**Do you want your identity to be public for this peer review?** For information about this choice, including consent withdrawal, please see our Privacy Policy .

Reviewer #1: No

---

## [Author Response · Author response to Decision Letter 0]

28 Oct 2024

Dear editor,

We thank the reviewer for their thoughtful reading, encouraging words and helpful feedback. We address all the points below.

1. Fundamental concepts and scope

“There appears to be a disjunct between the introduction and the rest of the manuscript; there are also some conceptual inconsistencies within the introduction.

# 1 #

First, the introduction presents one conception of overdiagnosis when there are a range of definitions or conceptualisations which have continued to evolve in recent years. Given that this paper is fundamentally about what HPs take overdiagnosis to be, suggest it is wise to start by recognising that the concept and its definition is contested even among experts (whether or not all experts recognise this contestation!).”

Author response:

The debate about the concept, the semantics, the scope, and the solutions for this contentious phenomenon is indeed still ongoing. Although this systematic review (SR) is deliberately limited to overdiagnosis in the context of screening, it also occurs in a clinical setting, and even when confined to the setting of screening, experts may differ in how they would define overdiagnosis

We added this notion to the introduction section.

Changed text in MS from Line 75

Further, the introduction states that overdiagnosed people “cannot benefit from this early diagnosis” (p.3 L57) and that treating a person with an overdiagnosed condition “cannot provide any benefit” p. 3 L65), but also states that the harms caused by overdiagnosis “must be weighed against the benefits of screening” (p.4 L82). The manuscript itself presents nuanced accounts of the complex benefit/harm balance involved in understanding and identifying overdiagnosis. Arguably this nuance is central to the main summary figure, Fig2. I would suggest this more nuanced approach is more consistent with the reality of what clinicians (and their patients) face and should be reflected in the introduction.

Author response:

We added some lines to the introduction to emphasise the difficult trade-off and uncertainty screening candidates and their HPs have to deal with when deciding about screening. We acknowledge that we emphasise the harms related to overdiagnosis. This is intentional, because this SR is not about HPs’ overall ideas about benefits and harms of screening, but deliberately zooms in on the (in our opinion) under-exposed problem of overdiagnosis.

Changed text in manuscript (line 95):

Finally, the manuscript uses the word screening without explaining how the study is conceptualising screening, or the imagined relationship between population screening and opportunistic testing. Your primary data seems to contain the views of both clinicians (who may be offering opportunistic tests to individuals, or may be informing individuals about population based screening programs) and administrators and others with an interest in population based programs. Suggest this screening-opportunistic testing distinction should be addressed directly, and you should make sure that when you report findings you are clear whether you are talking about a clinician (e.g. a GP) or an administrator (e.g. a program manager). This is especially pertinent to your third RQ which refers to ‘offering’ screening, but it is not completely clear what constitutes an offer in the context of the review and these different contexts.

Author response:

We thank the reviewer for drawing our attention to the fact that we did not convey this clearly in our SR. Indeed, we intend to encompass all forms of screening, because overdiagnosis is an issue in both population-based screening and opportunistic screening. We changed our manuscript at several locations to make this more clear (see below), and we also added a column in table 1 (overview of the included publications) to indicate whether the study took place in a setting with an ongoing population-based screening program or not.

Changed text in manuscript

Methods-section (Line 145)

Description of the included publications (Line 242)

2. Methods

As noted, the review is generally very strong.

There are reasons given for abandoning CIS in the discussion, but I wonder whether this information may be better in the methods? As there appears to be a significant amount of inductive analysis involved in the review, I was not 100% clear why CIS was abandoned. I was also not clear whether you would describe the methodology you followed as bespoke, or whether you found an alternate methodology? Some justification for why you did things the way you did them, if it was a bespoke method, would be helpful.

Author response

Thank you for this remark. We moved our explanation from the discussion to the methods section. After abandoning CIS, we did not literally follow a specific method . However, what we did aligns most with the method of Thematic Synthesis. We changed our methods section accordingly to convey our response to this remark.

Line 153

Can you say more about why you didn’t assess quality of the quantitative studies?

AUthor response

Thank you for drawing our attention that we did not make it sufficiently clear that we did assess the quality of the quantitative studies (see table 2). However we did not assess the certainty of evidence we obtained from these questionnaire studies because we could not find any formal grading tool to assess the certainty of evidence from questionnaire studies.

3. Findings

The knowledge/awareness results seemed thin, although this is arguably compensated by later sections.

The most substantive section is the ‘perception of overdiagnosis’ section. This section appears to rely heavily on a small number of the studies found based on the quotations used. If this is the case, suggest addressing this more directly.

Author response

We added to each section the number of studies our findings are based on.

The important ‘perceptions of overdiagnosis’ section also appears to rely heavily on quotes rather than presenting very much new analysis from the authors of this review. Can you provide any further analytic depth that draws out new insights by comparing & further synthesising the primary studies?

Author response:

We believe our main analysis lies in the fact that HPs’ perception of overdiagnosis and how they deal with it cannot be separated from their beliefs in the benefits of screening. This analysis covers the entirety of the results. We moved this overarching analysis to the beginning of the results section in order to highlight the new in depth insights.

MS Line 306

The quantitative and qualitative findings seem quite disconnected in the results section – is it possible to connect them any further?

Author response

Thank you for this suggestion. Throughout the manuscript we tried to connect both types of results more on a content level, while still maintaining our segregated analysis, as described in the methods section.

Figure 2 is useful and interesting and will be an important contribution of the study.

The discussion is a strength of the MS and raises some important and useful implications.

The authors want to thank the reviewer for this positive comment.

---

## [Editor Report · Decision Letter 1]

22 Nov 2024

Do health professionals know about overdiagnosis in screening, and how are they dealing with it? A mixed-methods systematic scoping review.

PONE-D-24-22152R1

Dear Dr. Piessens,

We’re pleased to inform you that your manuscript has been judged scientifically suitable for publication and will be formally accepted for publication once it meets all outstanding technical requirements.

Kind regards,

Pascal A. T. Baltzer, M.D.

Academic Editor

PLOS ONE
---

## [Editor Report · Acceptance letter]

PONE-D-24-22152R1

PLOS ONE

Dear Dr. Piessens,

I'm pleased to inform you that your manuscript has been deemed suitable for publication in PLOS ONE. Congratulations! Your manuscript is now being handed over to our production team.

Kind regards,

on behalf of

Dr. Pascal A. T. Baltzer

Academic Editor

PLOS ONE